# *Wolbachia* in *Aedes koreicus*: Rare Detections and Possible Implications

**DOI:** 10.3390/insects13020216

**Published:** 2022-02-21

**Authors:** Claudia Damiani, Alessia Cappelli, Francesco Comandatore, Fabrizio Montarsi, Aurelio Serrao, Alice Michelutti, Michela Bertola, Maria Vittoria Mancini, Irene Ricci, Claudio Bandi, Guido Favia

**Affiliations:** 1School of Biosciences and Veterinary Medicine, University of Camerino, CIRM Italian Malaria Network, Via Gentile III da Varano, 62032 Camerino, Italy; claudia.damiani@unicam.it (C.D.); alessia.cappelli@unicam.it (A.C.); aureliogiuseppe.serrao@unicam.it (A.S.); irene.ricci@unicam.it (I.R.); 2Biovecblok s.r.l., Via del Bastione 5, 62032 Camerino, Italy; 3Romeo ed Enrica Invernizzi Pediatric Research Center, Department of Biomedical and Clinical Sciences Luigi Sacco, Università di Milano, Via Giovanni Battista Grassi 74, 20157 Milan, Italy; francesco.comandatore@unimi.it (F.C.); claudio.bandi@unimi.it (C.B.); 4Istituto Zooprofilattico Sperimentale delle Venezie, Viale dell’Università 10, Legnaro, 35020 Padova, Italy; fmontarsi@izsvenezie.it (F.M.); amichelutti@izsvenezie.it (A.M.); mbertola@izsvenezie.it (M.B.); 5Polo d’Innovazione di Genomica, Genetica e Biologia, Via Mazzieri, 05100 Terni, Italy; mv.mancini@pologgb.com

**Keywords:** *Aedes koreicus*, microbiota, *Wolbachia*, *Asaia*

## Abstract

**Simple Summary:**

In Europe, the threat of emerging epidemic diseases also includes vector borne diseases. Several aedine mosquito species, showing vectorial capacity for pathogenic viruses, are invading, and expanding their ecological habitats. *Aedes albopictus*, a vector of Chikungunya, dengue, and Zika viruses, is well established in several European countries. In Italy, this species has been associated with public health issues, such as the Chikungunya outbreaks in 2007 and 2017, and by a dengue outbreak in 2020. Since 2011 a new invasive species, *Aedes koreicus,* has been recorded in Italy. The spreading of *Ae. koreicus* is presumably associated with passively spreading through the trading activity of specific goods among European countries, but multiple and independent introduction from native areas could be also possible. Given the risks associated with the presence of this vector, innovative and effective control measures are necessary. Recently, novel tools based on the use of symbiotic bacteria for the control of mosquito vectors have been proposed, focusing on *Asaia* and *Wolbachia* bacteria. Here, we report the first evidence of the presence of *Wolbachia* in a population of *Ae. koreicus*, cohabited by *Asaia* bacteria. These results open interesting prospects for the control of these invasive species.

**Abstract:**

The emerging distribution of new alien mosquito species was recently described in Europe. In addition to the invasion of *Aedes albopictus*, several studies have focused on monitoring and controlling other invasive *Aedes* species, as *Aedes koreicus* and *Aedes japonicus*. Considering the increasing development of insecticide resistance in *Aedes* mosquitoes, new control strategies, including the use of bacterial host symbionts, are proposed. However, little is known about the bacterial communities associated with these species, thus the identification of possible candidates for Symbiotic Control is currently limited. The characterization of the natural microbiota of field-collected *Ae. koreicus* mosquitoes from North-East Italy through PCR screening, identified native infections of *Wolbachia* in this species that is also largely colonized by *Asaia* bacteria. Since *Asaia* and *Wolbachia* are proposed as novel tools for Symbiotic Control, our study supports their use for innovative control strategies against new invasive species. Although the presence of *Asaia* was previously characterized in *Ae. koreicus*, our study characterized this *Wolbachia* strain, also inferring its phylogenetic position. The co-presence of *Wolbachia* and *Asaia* may provide additional information about microbial competition in mosquito, and to select suitable phenotypes for the suppression of pathogen transmission and for the manipulation of host reproduction in *Ae. koreicus*.

## 1. Introduction

Currently, five *Aedes* Invasive Mosquito species are known to be established in Europe [1]. Among these, *Aedes albopictus* is certainly the most widespread mosquito vector of arboviruses in Europe, including dengue, Chikungunya, and Zika viruses. The other *Aedes* species present in Europe are *Aedes japonicus* and *Aedes koreicus*, while *Aedes atropalpus* and *Aedes aegypti* are occasionally detected [2].

Several autochthonous cases of Chikungunya and dengue in Italy, France, Croatia, and Portugal have recently been documented [3,4]. Due to wide distribution and abundance of *Ae. albopictus* in Italy, there were some outbreaks of Chikungunya, the most relevant of which was in 2007 in the Ravenna province where more than 200 cases were recorded [5]. After this, the control of *Ae. albopictus* in the Emilia-Romagna region involved about 300 municipalities, with 4.2 million inhabitants. Public costs for this intervention in 2008–2011 was estimated in approximately EUR 5.5 million/year. In the last few years (2017) new autochthonous outbreaks of Chikungunya occurred in Lazio and Calabria regions and in 2020 the first autochthonous outbreaks of dengue occurred in the Veneto region [4,6,7].

In the last decade, *Ae. koreicus* (Edwards, 1917) and *Ae. japonicus* (Theobald, 1901), have emerged and now have become permanently established in some areas of northern Italy [8,9]. Considering that these two species are potential vectors of important arboviruses, their distribution and settlement in new areas may represent a threat to human and animal health. The vector competence *Ae. koreicus* is still unclear, however the capacity to transmit Japanese Encephalitis Virus [10], Chikungunya virus [11], and *Dirofilaria immitis* has been documented [12].

Several *Aedes* invasive mosquitoes are characterized by high ability to spread, facilitated by human transportation, and to adapt to new environmental conditions. Considering their capacity to transmit pathogens of human and animal health interest, innovative control methods are urgently needed. Recently, alternative strategies for vector control investigated the use of microorganisms, which collectively take the name of Symbiotic Control, representing a multifaceted approach using basically: (i) the disruption of microbial symbionts required by insect pests; (ii) the manipulation of symbionts that can express anti-pathogen molecules within the host; and (iii) the introduction of endogenous microbes that affect life-span and vector capacity of the new hosts in insect populations [13].

The identification of microbial symbionts to be enrolled in the Symbiotic Control of mosquito borne diseases is, therefore, a fundamental prerequisite. Consequently, we performed a study aimed to characterize the microbiota of an Italian population of *Ae. koreicus* with particular interest in the identification of two bacterial symbionts, *Wolbachia* and *Asaia*, that have been proved to be useful in Symbiotic Control applications as shown, respectively, by the “eliminate dengue program” and by some laboratory paratransgenic practices in malaria vectors [14,15]. Here, we report the identification of both symbionts, while the presence of *Asaia* strains was previously characterized in *Ae. koreicus*, this study offers a characterization of this *Wolbachia* strain, while inferring its phylogenetic position.

## 2. Materials and Methods

### 2.1. Mosquito Collection, Identification and Rearing

Mosquitoes were collected as part of the *Aedes* Invasive Mosquito monitoring program, realized by the Parasitology laboratory of the Istituto Zooprofilattico Sperimentale delle Venezie (IZSVe) since 2011 and implemented every year. Mosquitoes were mainly collected from artificial containers by dipping and delivered to the laboratory in a box fridge. Mosquitoes were morphologically identified according to Montarsi et al., 2013 [16], and molecularly as described in Schneider et al., 2016 [17], as control we used reference DNAs from *Ae. koreicus*, *Ae. aegypti,* and *Ae. albopictus*.

In 2019, *Ae. koreicus* larvae were collected at Val di Zoldo (N 46,345813; E 12,188762; Belluno province, North-East Italy) and then reared for one generation in the insectary of the University of Camerino under standard laboratory conditions (29 °C and 85% ± 5 relative humidity) and photoperiods (12:12 light–dark). Insects were maintained on a sterile 10% sucrose solution ad libitum. Adult female mosquitoes were collected for a preliminary microbiota investigation by 16S Miseq analysis. In 2019 and 2020, a total of 148 *Ae. koreicus* were collected in several villages in North-East of Italy (Appendix A) for a deep investigation of the presence of the bacteria *Asaia* and *Wolbachia*. 

### 2.2. DNA Extraction

Before the DNA extraction, the insect surface was sterilized in 70% ethanol and rinsed for three times in sterile PBS. Samples were homogenized with sterile 0.5-mm wide glass beads (Bertin Instrument, Montigny-le-Bretonneux, France) for 30 s at 6800 rpm by automatic tissue homogenizer (Precellys 24, Bertin Instrument, Montigny-le-Bretonneux, France). Genomic DNA was extracted using a JetFlex Genomic DNA Purification kit (Invitrogen, Thermo Fisher Scientific, Waltham, MA, USA) according to the manufacturer’s instructions.

### 2.3. Metagenomics

For a preliminary investigation of *Ae. koreicus* microbiota, six female mosquitoes were analyzed by 16S next generation sequencing. 16S sequencing analysis was conducted by LGC Genomics (Berlin, Germany). Libraries preparation was performed by amplifying the hypervariable region V3–V4 of 16S ribosomal RNA using 341F and 785R oligonucleotides [18]. Data were pre-processed using the Illumina bcl2fastq 2.17.1.14 software and reads sorted by amplicon inline barcodes. Sequencing adapter remnants were clipped from all reads. 16S pre-processing and OTU picking from amplicons were analyzed using Mothur 1.35.1 [19]. The sequence alignments were performed against the 16S Mothur-Silva SEED r119 reference alignment. OTU diversity was analyzed with QIIME 1.9.0 [20] and annotations of putative species level of OTUs were obtained with NCBI BLAST+ 2.2.29 [21]. A negative control consisting of a blank sample was included for each batch of extraction to control for contamination of bacteria possibly introduced during the DNA extraction. They were not further processed since no quantifiable extract was produced from each negative control. All the reads related to the 16S Miseq analysis have been deposited in the EMBL Nucleotide Sequence Database (NCBI) with the accession number (Bioproject PRJNA673771).

### 2.4. Wolbachia and Asaia Detection

PCRs for the detection of *Asaia* and *Wolbachia* were performed using specific oligonucleotides. For *Wolbachia*, a semi-nested PCR targeting on the *Wolbachia* 16S was performed using 50 ng genomic DNA, 1X Buffer, 0.25 mM dNTPs, 0.9U DreamTaq Polymerase (Thermo Scientific, Waltham, MA, USA), 240 nM Wol-For, 160 nM Wol-rev2 and 120 nM Wol-rev3 [22]. The amplification cycle consisted of an initial denaturation at 95 °C for 3 min was followed by 5 cycles consisting of denaturation at 95 °C for 30 s, annealing at 54 °C for 30 s, and extension at 72 °C for 30 s, and 25 cycles consisting of denaturation at 95 °C for 30 s, annealing at 52 °C for 30 s, and extension at 72 °C for 30 s, concluding with a final extension step of 10 min at 72 °C. For *Asaia*, specific PCR targeting the 16S gene was performed using 50 ng genomic DNA, 1X Buffer, 0.25 mM dNTPs, 0.9U DreamTaq Polymerase (Thermo Scientific, Waltham, MA, USA), 200 nM of AsaiaNewFor and AsaiaNewRev oligonucleotides [23]. The amplification protocol included: an initial denaturation at 95 °C for 3 min, followed by 30 cycles consisting of denaturation at 95 °C for 30 s, annealing at 60 °C for 30 s, and extension at 72 °C for 30 s, concluding with a final extension step of 10 min at 72 °C. The PCR products were electrophoresed on a 1% agarose gel to determine the presence and size of the amplified DNA. The amplicons were purified and sequenced. The 16S RNA sequence were analyzed by BLASTN (http://blast.ncbi.nlm.nih.gov/Blast.cgi accessed on 27 July 2020). Sequence alignment was then generated by ClustalW (http://www.genome.jp/tools/clustalw/ accessed on 27 July 2020). Finally, the sequence of the 16S gene of *Wolbachia* was deposited through the EMBL-Bank (Accession number MT809043).

### 2.5. Molecular and Phylogenetic Wolbachia Characterization

The five genes of the *Wolbachia* Multi-Locus Sequence Typing (MSLT) scheme (gatB, coxA, hcpA, ftsZ, and fbpA) were amplified, as described in Baldo et al. 2006 [24] (https://pubmlst.org/organisms/wolbachia-spp/protocol-single-infected accessed on 19 December 2021) using DreamTaq Polymerase (Thermo Scientific, Waltham, MA, USA) protocol. The PCR products were purified, and Sanger sequenced. The resulting sequences were used for the phylogenetic analysis as follows: a dataset of 49 complete genomes spanning the genetic variability of *Wolbachia* were manually retrieved from the PATRIC database [25] and, for each genome, the five MLST scheme genes were extracted by BLAST search. For each of the five MLST scheme genes, the 49 sequences from the background genomes and the two from the strains of this study were aligned using Muscle [26] and then concatenated. Maximum Likelihood (ML) phylogenetic analysis with 100 pseudo-bootstraps was then performed on the aligned genes concatenate using RAxML8 [27], setting the GTR+I+G model, previously selected using model test-ng tool [28]. All the reads related to the MLST sequences described in this study have been deposited in the European Nucleotide Archive (ENA) (FbpA BankIt2532145 OL960727; FtsZ BankIt2532145 OL960728; HcpA BankIt2532145 OL960729; CoxA BankIt2532145 OL960730; GatB BankIt2532145 OL960731).

## 3. Results and Discussion

*Aedes koreicus* adult mosquitoes generated from field collected larvae, were analyzed by 16S metagenomics analysis revealing that *Asaia* was the most abundant symbionts (Appendix A). *Asaia* is a very well-characterized symbiont of mosquito and other insects [23,29] and it has been proposed as a potential control tool for mosquito-borne diseases for direct paratransgenic applications and indirectly through the upregulation of the host immune response [13,14,15,16,17,18,19,20,21,22,23,24,25,26,27,28,29,30]. Since in some mosquito populations, *Asaia* seems to outcompete with *Wolbachia* symbiont [22], we decided to specifically verify the presence of these two symbionts in a larger number of adults of *Ae. koreicus* sampled in several villages in North-East of Italy in some of those regions in which this invasive species is predominantly localized (Figure 1).

Although the COVID-19 pandemic posed serious constraints in the new collection campaigns, 148 samples were collected (Appendix A). Although *Asaia* was detected in 126 of the 148 individuals examined, in a few of them (2/148) we also detected *Wolbachia* (Table 1). 

This finding is of particular interest since *Wolbachia* is a symbiont already successfully used in the control of mosquito-borne diseases, in particular dengue [31], and has never been documented prior to this study in *Ae. koreicus*. The finding of *Wolbachia* in *Ae. koreicus* pushed us to a molecular-genetic characterization of this bacterium. Consequently, the *Wolbachia* bacteria from the two positive individual strains were characterized by the sequencing of the five genes of the Multi-Locus Sequence Typing (MLST) *Wolbachia* scheme and successive phylogenetic analysis. The resulting tree, reported in Figure 2, clearly shows that the two strains are genetically very similar and belong to the Supergroup B, and are closely related to the wAlbB strain, one of the two native strains of *Ae. albopictus* [32].

The ability of some *Wolbachia* strains to reduce the lifespan of *Ae. aegypti*, to invade populations of mosquitoes through the induction of the Cytoplasmic Incompatibility phenomenon and to interfere with the development of a variety of pathogens, has “identified” this bacterium as an effective innovative tool for the control of mosquito-borne diseases [33,34]. In this context, the first description of the association between *Wolbachia* and *Ae. koreicus* goes far beyond the pure descriptive connotation but opens interesting perspectives for investigations aimed at the detection and characterization of *Wolbachia* in other *Ae. koreicus* populations, as well as of other bacterial symbionts, such as the acetic bacterium *Asaia*.

## 4. Conclusions

Although based on a limited set of data due to the inherent pilot nature of this study, the differential distribution of *Asaia* and *Wolbachia* in *Ae. koreicus* population seems coherent with the mutual exclusion between the two symbionts, as previously highlighted in other mosquito populations [22], thus providing additional information about microbial competition in mosquito. All this complements the characterization of the first *Wolbachia* strain described in *Ae. koreicus,* likely offering an opportunity to select suitable phenotypes for the suppression of pathogen transmission and for the manipulation of host reproduction in this invasive species, as well as in other mosquito vectors. Moreover, new field collections are planned in the same areas of the previous samplings and in neighboring areas to expand the sample size and to further identify native *Wolbachia* strains circulating within *Ae. koreicus* population. The selection of *Wolbachia*-carrying females will allow the establishment of a laboratory colony of *Ae. koreicus*, representing an essential step for a robust characterization of symbiont–host interactions. Consequently, if showing suitable promising traits, the newly identified *Wolbachia* strains could be likely harnessed in different populations of mosquito for developing *Wolbachia*-based control interventions.

## Figures and Tables

**Figure 1 insects-13-00216-f001:**
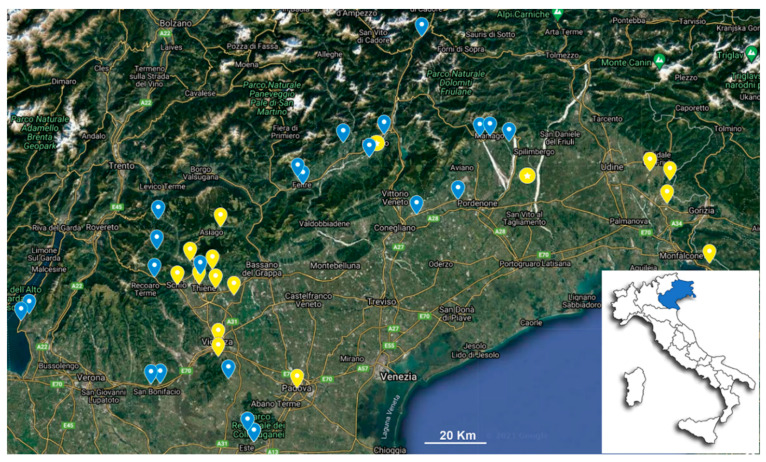
Description map of mosquito collection sites. The collection sites of 2019 and 2020 is represented in yellow and blue, respectively. The circle with star indicates the site where *Wolbachia* is detected. The map was built using google maps application.

**Figure 2 insects-13-00216-f002:**
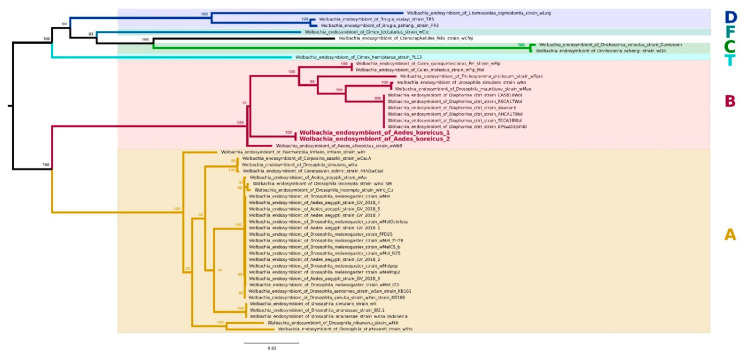
Multi-locus *Wolbachia* tree. Maximum Likelihood (ML) phylogenetic tree obtained by analyzing the concatenate of the Multi-Locus Sequence Typing (MLST) genes. The names of the two *Wolbachia* strains isolated in this work are bolded and colored in red on the tree. The *Wolbachia* supergroup monophyla are highlighted on the tree by branch colors and colored squares: supergroup A in yellow, B in red, T in azure, C in green, F in blue, and D in violet. The ML bootstrap support values are reported on the relative nodes.

**Table 1 insects-13-00216-t001:** Percentage of positivity for *Asaia* and *Wolbachia*.

	*Asaia*	*Wolbachia*
	positive/total (%)	positive/total (%)
*Aedes koreicus females*	66/85 (76.6)	2/85 (2.6)
*Aedes koreicus males*	60/63 (95.2)	0/63 (0)
Total	126/148 (85.1)	2/148 (1.4)

## Data Availability

All the reads related to the 16S Miseq analysis (Bioproject PRJNA673771) and the *Wolbachia* 16S RNA sequence (accession number MT809043) have been deposited in The EMBL Nucleotide Sequence Database (NCBI). All the reads related to the MLST sequences described in this study have been deposited in the European Nucleotide Archive (ENA) (FbpA BankIt2532145 OL960727; FtsZ BankIt2532145 OL960728; HcpA BankIt2532145 OL960729; CoxA BankIt2532145 OL960730; GatB BankIt2532145 OL960731.

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
