# Peer review of "Wolbachia* in *Aedes koreicus*: Rare Detections and Possible Implications"

_insects, 2022, doi:10.3390/insects13020216_

Round 1
Reviewer 1 Report
This is a useful manuscript describing the presence of Wolbachia and Asaia bacteria in invasive populations of Ae. koreicus in Italy — these findings are relevant to the development of control strategies for mosquito-borne disease. Overall, I find the manuscript to be clearly written and the findings to be robustly supported — for this reason, I am happy to recommend publication without further revision (although I have noted a couple of minor tweaks and typos below).
minor comments:
L75: readers with less familiarity with Symbiotic Control might appreciate a sentence or two and a couple citations explaining how this strategy works.
L108: “newt” should be “next”?
Author Response
Replies to Reviewer 1
This is a useful manuscript describing the presence of Wolbachia and Asaia bacteria in invasive populations of Ae. koreicus in Italy — these findings are relevant to the development of control strategies for mosquito-borne disease. Overall, I find the manuscript to be clearly written and the findings to be robustly supported — for this reason, I am happy to recommend publication without further revision (although I have noted a couple of minor tweaks and typos below).
Reply) We really thank Reviewer 1 for the very positive review and useful suggestions.
minor comments:
L75: readers with less familiarity with Symbiotic Control might appreciate a sentence or two and a couple citations explaining how this strategy works.
Reply) Based on the appropriate suggestion, we have inserted an articulate sentence describing the characteristics of the symbiotic control and how it works (Lines 77-82).
L108: “newt” should be “next”?
Reply) We have modified the manuscript accordingly
Reviewer 2 Report
General comments
Damiani et al. surveyed Italian populations of Aedes koreicus, an invasive mosquito species, for the bacterial symbionts Asaia and Wolbachia. They find a high frequency of Asaia infection, and interestingly, two individuals that were positive for Wolbachia. Strains of Wolbachia are being used successfully to control Aedes aegypti and Ae. albopictus mosquito populations, so understanding the infection status of invasive Aedes populations is important for future control efforts. This study is the first report of Wolbachia in Ae. koreicus. Although this is an important finding, the authors should emphasize that this work is preliminary. They use PCR and MLST markers to detect Wolbachia and place the strain in a phylogeny with related strains, but confirming that an insect species carries an active Wolbachia infection requires further validation (see Chrostek and Gerth 2019 mBio). The authors should consider including a section outlining potential future work to 1. further validate and characterize the strain, and 2. Harness this strain to control mosquito populations
Specific comments
Abstract and introduction - “associated to” should be “associated with”
Line 49 – the acronym for Aedes Invasive Mosquito (AIM) is mentioned infrequently in the manuscript- consider writing this out in full each time because it can be hard to follow
Line 50 – change “present and spread” to “widespread”
Line 75- It is inappropriate to use an acronym here as SC is only used once in the manuscript and is not in common usage.
Line 75 – Refer to previous studies that have used endosymbionts to control mosquito-borne diseases. E.g. population replacement of Ae. aegypti with Wolbachia in several countries, and suppression of various Aedes and Culex species through cytoplasmic incompatibility
Line 79- Give examples of studies that have used Wolbachia and Asaia to control mosquitoes.
Line 91 – please provide specific details if possible, including any relevant controls.
Line 108 – “newt generation sequencing” should be “next generation sequencing”
Table 1 – “pos/tot” should be written out in full: “positive/total”
Lines 234 – Wolbachia has not just been “identified”- it is already being used as an effective tool for dengue control. The authors should also refer to studies which demonstrate effects of Wolbachia on mosquito longevity (e.g. McMeniman et al. 2009 Science) and pathogen blocking (Moreira et al. 2009 Cell)
Line 242- were the two individuals that tested positive for Wolbachia also positive for Asaia? If so then this would be evidence against the mutual exclusion hypothesis
Results – was there a geographic pattern of Asaia infection, or was it random? Is the infection frequency reported here consistent with previous work?
Author Response
Replies to Reviewer 2
Damiani et al. surveyed Italian populations of Aedes koreicus, an invasive mosquito species, for the bacterial symbionts Asaia and Wolbachia. They find a high frequency of Asaia infection, and interestingly, two individuals that were positive for Wolbachia. Strains of Wolbachia are being used successfully to control Aedes aegypti and Ae. albopictus mosquito populations, so understanding the infection status of invasive Aedes populations is important for future control efforts. This study is the first report of Wolbachia in Ae. koreicus. Although this is an important finding, the authors should emphasize that this work is preliminary. They use PCR and MLST markers to detect Wolbachia and place the strain in a phylogeny with related strains, but confirming that an insect species carries an active Wolbachia infection requires further validation (see Chrostek and Gerth 2019 mBio). The authors should consider including a section outlining potential future work to 1. further validate and characterize the strain, and 2. Harness this strain to control mosquito populations
Reply) We thank Reviewer 2 for the positive review and for the very useful suggestions. We have further stressed that this is a pilot study and that the results are preliminary. Moreover, we have improved the conclusion section indicating future work to validate and characterize the Wolbachia strain and how to harness it to control mosquito populations.
Specific comments
Abstract and introduction - “associated to” should be “associated with”
Reply) We modified the text accordingly
Line 49 – the acronym for Aedes Invasive Mosquito (AIM) is mentioned infrequently in the manuscript- consider writing this out in full each time because it can be hard to follow
Reply) We modified the text accordingly
Line 50 – change “present and spread” to “widespread”
Reply) We modified the text accordingly
Line 75- It is inappropriate to use an acronym here as SC is only used once in the manuscript and is not in common usage.
Reply) We modified the text accordingly
Line 75 – Refer to previous studies that have used endosymbionts to control mosquito-borne diseases. E.g. population replacement of Ae. aegypti with Wolbachia in several countries, and suppression of various Aedes and Culex species through cytoplasmic incompatibility
and
Line 79- Give examples of studies that have used Wolbachia and Asaia to control mosquitoes.
Reply) Consistent with both these appropriate suggestions of the reviewer, we have provided some examples of control applications and we modified the text inserting specific references to previous experiences of Wolbachia-based dengue control and paratransgenesis in malaria vectors (lines 77-82 and lines 87-89).
Line 91 – please provide specific details if possible, including any relevant controls.
Reply) We have specified that we have used as controls DNA from Ae. koreicus, Ae. aegypti and Ae. albopictus (Lines 124-125)
Line 108 – “newt generation sequencing” should be “next generation sequencing”
Reply) We modified the text accordingly
Table 1 – “pos/tot” should be written out in full: “positive/total”
Reply) We modified text accordingly
Lines 234 – Wolbachia has not just been “identified”- it is already being used as an effective tool for dengue control. The authors should also refer to studies which demonstrate effects of Wolbachia on mosquito longevity (e.g. McMeniman et al. 2009 Science) and pathogen blocking (Moreira et al. 2009 Cell)
Reply) We referred to these studies adding in the bibliography the appropriate references (Lines 278-279)
Line 242- were the two individuals that tested positive for Wolbachia also positive for Asaia? If so then this would be evidence against the mutual exclusion hypothesis
Reply) This aspect is certainly critical. The analysis of the samples showed that Asaia is present in the different mosquitoes analyzed in different quantities. The two mosquitoes that also tested positive for Wolbachia have fairly low levels of Asaia. However, given the small number of Wolbachia-positive samples, at the moment it does not seem possible to affirm that this confirms a relationship of mutual exclusion between Asaia and Wolbachia in Ae. koreicus.
Results – was there a geographic pattern of Asaia infection, or was it random? Is the infection frequency reported here consistent with previous work?
Reply) The geographic pattern is random. The infection frequency is very consistent with that reported in previous work.